# Burden of disease and productivity impact of *Streptococcus suis* infection in Thailand

**Ajaree Rayanakorn** [1,2] *, **Zanfina Ademi** [3], **Danny Liew** [3], **Learn-Han Lee** [1] *

**1** Novel Bacteria and Drug Discovery Research Group (NBDD), Microbiome and Bioresource Research Strength, Jeffrey Cheah School of Medicine and Health Sciences, Monash University Malaysia, Bandar Sunway, Malaysia, **2** Faculty of Public Health, Chiang Mai University, Chiang Mai, Thailand, **3** School of Public Health and Preventive Medicine, Monash University, Melbourne, Australia

* ajaree.rayanakorn@cmu.ac.th, ajaree.rayanakorn@monash.edu (AR); lee.learn.han@monash.edu (L-HL)

**Data Availability Statement:** All relevant data are within the manuscript and its Supporting Information files.

**Funding:** The author(s) received no specific funding for this work.

## Abstract

### Background

*Streptoccocus suis (S.suis)* infection is a neglected zoonosis disease in humans mainly affects men of working age. We estimated the health and economic burden of *S.suis* infection in Thailand in terms of years of life lost, quality-adjusted life years (QALYs) lost, and productivity-adjusted life years (PALYs) lost which is a novel measure that adjusts years of life lived for productivity loss attributable to disease.

### Methods

A decision-analytic Markov model was developed to simulate the impact of *S. suis* infection and its major complications: death, meningitis and infective endocarditis among Thai people in 2019 with starting age of 51 years. Transition probabilities, and inputs pertaining to costs, utilities and productivity impairment associated with long-term complications were derived from published sources. A lifetime time horizon with follow-up until death or age 100 years was adopted. The simulation was repeated assuming that the cohort had not been infected with *S.suis*. The differences between the two set of model outputs in years of life, QALYs, and PALYs lived reflected the impact *of S.suis* infection. An annual discount rate of 3% was applied to both costs and outcomes. One-way sensitivity analyses and Monte Carlo simulation modeling technique using 10,000 iterations were performed to assess the impact of uncertainty in the model.

### Key results

This cohort incurred 769 (95% uncertainty interval [UI]: 695 to 841) years of life lost (14% of predicted years of life lived if infection had not occurred), 826 (95% UI: 588 to 1,098) QALYs lost (21%) and 793 (95%UI: 717 to 867) PALYs (15%) lost. These equated to an average of 2.46 years of life, 2.64 QALYs and 2.54 PALYs lost per person. The loss in PALYs was associated with a loss of 346 (95% UI: 240 to 461) million Thai baht (US$11.3 million) in GDP, which equated to 1.1 million Thai baht (US$ 36,033) lost per person.

**Competing interests:** The authors have declared that no competing interests exist.

## Conclusions

*S.suis* infection imposes a significant economic burden both in terms of health and productivity. Further research to investigate the effectiveness of public health awareness programs and disease control interventions should be mandated to provide a clearer picture for decision making in public health strategies and resource allocations.

## Author summary

*Streptoccocus suis (S.suis)* infection is a potentially lethal zoonotic disease in humans. In the present study, we sought to estimate the impact of the disease in Thailand in terms of years of life lost, quality-adjusted life years (QALYs) lost, and productivity-adjusted life years (PALYs) lost. A decision-analytic Markov model was developed to simulate the impact of *S.suis* infection and its major complications among Thai people. In 2019, it was estimated that the infection incurred 769 years of life lost (14% of predicted years of life lived if infection had not occurred), 826 QALYs lost (21%) and 793 PALYs (15%) lost. These equated to an average of 2.5 years of life, 2.6 QALYs and 2.5 PALYs lost per person. The loss in PALYs was associated with a loss of 346 million Thai baht (US$11.3 million) in GDP, which equated to 1.1 million Thai baht (US$ 36,033) lost per person. The findings call for increased public health awareness and comprehensive efforts to control and prevent the disease.

## Introduction

*Streptococcus suis (S.suis)* is a gram positive alpha-hemolytic bacteria whose natural host is usually the pig. It can cause serious infection in humans through contact with pigs or pig meat, especially via the ingestion of uncooked pork. Meningitis (68%), sepsis (25%) and infective endocarditis (12.4%) are major clinical manifestations of *S.suis* infection, with a case fatality of 13% [1]. Long-term complications among survivors comprise sensorineural hearing loss (SNHL) with or without vestibular dysfunction and valvular heart disease. SNHL is the most common complication among *S.suis* meningitis patients which is typically bilateral and permanent [1,2]. The highest prevalence of *S.suis* infection is in South East Asia, notably Thailand (0.487 per 100,000) [3] and Vietnam (0.249 per 100,000) [4]. Globally, there are more than 1,600 reported cases [1]. The disease predominantly affects males from young adulthood to middle age [1,5]. This may be due to their risk behaviors including raw pork consumption, exposure to pigs or raw pork through their occupations and slaughtering activity [5]. While the disease mainly affects farmers or abattoir workers in western countries, it was found that ingestion of raw or undercooked pork is an important risk factor in Asia, especially in Thailand and Vietnam, where raw pork consumption is traditionally practiced.

*S.suis* infection has been recognised as the cause of substantial loss in swine industry [6]. The only previous study on economic burden from *S.suis* infection in human was conducted in Vietnam [4]. In this study, the burden of disease was quantified in term of disability-adjusted life years (DALYs). The DALYs lost ranged 1,437–1,866 from 2011–2014 with the mean direct cost per episode of US$1,635 (95%CI 1,352–1,923) reflecting a significant economic impact [4]. However, long-term consequences including cardiac complication after infective endocarditis (IE) and follow-up audiological testing were not captured in the model.

The fact that the impact of *S.suis* infection is beyond acute infection involving long-term impairments including SNHL, valvular heart disease and premature mortality is of paramount importance to address the long-term sequelae of this sophisticated neglected infection.

To our knowledge, no previous research has estimated the health and economic burden of *S.suis* infection in Thailand. Our study aimed to quantify this burden in terms of years of life, quality-adjusted life years (QALYs), and productivity-adjusted life years (PALYs) lost, as well as impact on gross domestic product (GDP).

## Methods

### Model description

The model comprised two parts (Fig 1). The first was a decision analytic tree which reflected the first 30 days of *S.suis* infection, over which time subjects could: i) completely recover without any long-term sequelae; ii) survive meningitis (with or without other complications); iii) survive infective endocarditis (IE, with or without other complications except meningitis); or iv) die. The second part of the model consisted of a Markov model with seven long-term health states: i) completely recovered; ii) post meningitis without complications; iii) post meningitis with long-term hearing loss but no vestibular dysfunction; iv) post meningitis with long-term hearing loss and vestibular dysfunction; v) post IE without long-term complications; vi) post IE with long-term complications; and vii) dead. Vestibular dysfunction without hearing loss and endopthalmitis are rare, and therefore were not considered. Cycle lengths in the Markov model were one year.

For all the living health states, only two transition states were considered: stay alive and die. We assumed that those who survived meningitis, regardless of whether or not they had hearing loss or vestibular dysfunction, had the same annual risks of dying as the sex-and-age matched general population. The same assumption was applied to those who suffered IE without long-

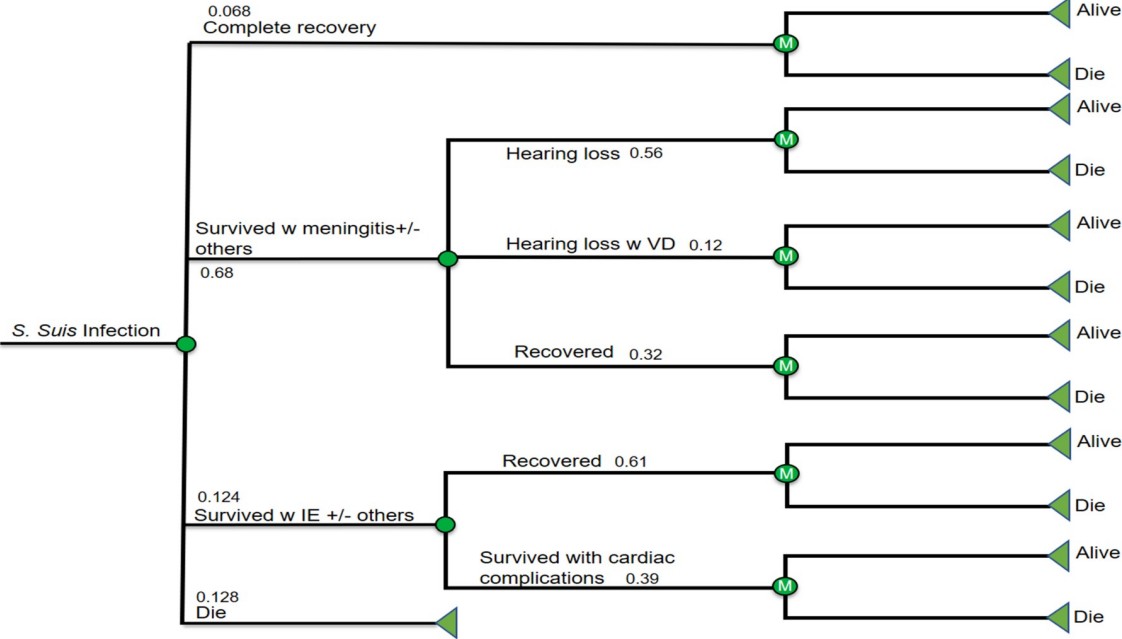

**Fig 1. A decision tree and Markov models.** The infection involves acute phase and post-infection phase in which there are different health states from complete recovery, partial recovery (hearing loss, hearing loss w VD, not recovered from IE) and death.

term complications. By contrast, those who suffered IE with long-term complications were 2.2 times more likely to die compared to the general population [7].

The model population comprised 312 Thais (239 males and 73 females) predicted to develop *S.suis* infection in 2019. The baseline age was assumed to be 51 years, which is the mean age of onset of acute infection [1]. The cohort was stratified by sex and followed up until age 100 years, which means the time horizon was 49 years. An annual discount rate of 3% was applied to both costs and outcomes in accordance with Thai Health Technology Assessment (HTA) guidelines [8].

To estimate the impact of the disease, the model simulation was repeated assuming that the cohort did not contract *S.suis* infection. Differences in the results of these two simulations in terms of years of life, QALYs and PALYs reflected the burden attributable to *S.suis* infection. PALYs are a novel measure that adjust years of life lived for productivity loss attributable to the disease in the same way that QALYs adjust years if life lived for impaired quality of life [9].

## Data sources

### Epidemiological data

Key input parameters and their sources are summarized in Table 1.

Demographic and mortality data for the general Thai population were obtained from the Country Office of Insurance Commission [10]. Predicted incident cases of *S.suis* infection in 2019 (N = 312) was estimated based on the annual incidence data for 2015–2018 reported by the Thailand Bureau of Epidemiology at the Ministry of Public Health [3]. The proportion of males (76.6%) was based on a meta-analysis [4]. Case fatality during the acute infection and likelihoods of developing meningitis and IE were derived from a systematic review and meta-analysis [1]. Among subjects who developed meningitis, the probability of long-term hearing loss with and without vestibular dysfunction were obtained from a retrospective study in Northern Thailand [11]. Among subjects who developed IE, the probability of long-term cardiac complications was based on a systematic review and meta-analysis [12]. The relative risk of death among subjects with post IE patients with long-term cardiac complications compared to general population (2.2) was based on a Swedish cohort study [7]. No equivalent Thai data were available.

### Cost parameters

Treatment costs for the acute *S.suis* infection were based on average inpatient expenses per *S. suis* episode at Chiang Mai University Hospital, Northern Thailand from 2005 to 2018. These unpublished data were provided by the hospital's office of medical records and statistics and based on 133 consecutive patients (S1 Table). Local cost data on IE or rheumatic heart disease are lacking. Therefore, data from Thai Acute Coronary Syndrome (ACS) registry which is considered to be the most comprehensive cost study in Thailand was used to estimate chronic treatment costs of IE [14]. The cost for non-fatal myocardial infarction (MI) one month after the event for the first year and the following years were used as surrogate costs of IE. The cost of hearing aids were based on the maximum reimbursement rate per device under universal health coverage scheme according to the Thailand National Health Security Office [15]. The costs for routine hearing test and computed tomography (CT) of temporal bone were obtained from Thailand's standard costing list [22]. The number of tests conducted during follow-up was based on a case series of 40 adult *S.suis* infected patients in northern Thailand [16]. Cost data were inflated to year 2019 values applying consumer price index for medical care [23]. The average market exchange rate in Quarter 3, 2019 was $US1 = 30.7123 Thai baht (THB) [24].

**Table 1. Key input parameters.**

| Parameters | Base-case value | Ranges | Distributions | Source |
|---|---|---|---|---|
| Total population of Thailand in 2018 (million) | 68.416 | - | Fixed | Thailand Board of Investment. Thailand in Brief: Demographic. [13] |
| Annual disease incidence 2019 (per 100,000 population) | 0.457 | 0.420–0.493 | Fixed | Bureau of Epidemiology, Department of Disease Control, MoPH, Thailand*.[3] |
| Age of onset of acute infection (years) | 51.4 | 49.5–53.2 | Fixed | A systematic review and meta-analysis.[1] |
| **Transitional probabilities** | | | | |
| Case fatality rate | 0.128 | 0.090–0.180 | Uniform | A systematic review and meta-analysis.[1] |
| Infective endocarditis | 0.124 | 0.067–0.219 | Uniform | A systematic review and meta-analysis.[1] |
| Meningitis | 0.680 | 0.589–0.758 | Uniform | A systematic review and meta-analysis.[1] |
| Meningitis-hearing loss | 0.381 | 0.310–0.478 | Uniform | Retrospective cohort study [11]. |
| **Markov** | | | | |
| Probability of developing hearing loss | 0.680 | 0.544–0.816 | Uniform | Retrospective cohort study [11]. |
| Probability of developing hearing loss with VD | 0.120 | 0.096–0.144 | Uniform | Retrospective cohort study [11]. |
| Probability of developing complications from IE | 0.390 | 0.320–0.460 | Uniform | A systematic review and meta-analysis.[12] |
| Probability of dying after IE | 2.20 | 2.00–2.30 | Uniform | A nationwide cohort study.[7] |
| ***Acute treatment cost**** | | | | |
| IPD per episode ($US) | 124,675 (4,132) | 11,337–40,4863 (376–13,418) | Gamma | Office of medical records and statistics, CMU Hospital [S1 Table] |
| ***Chronic treatment cost*** | | | | |
| IE after 1 month within 1st year ($US) | 55,785 (1,849) | 44,628–66,942 (1,479–2,219) | Gamma | Thai Acute Coronary Syndrome (ACS) registry.[14] |
| IE following years ($US) | 15,934 (528) | 12,748–19,121 (422–634) | Gamma | Thai Acute Coronary Syndrome (ACS) registry.[14] |
| Hearing aids reimbursement ($US) | 12,769 (423) | 10,215–15,323 (339–508) | Gamma | National Health Security Office, Thailand.[15] |
| Audiometry ($US) | 220 (7.29) | 176–264 (6–9) | Gamma | Case series and Standard Cost List for Health Economic evaluation in Thailand.[8,16] |
| CT Temporal bone ($US) | 7,344 (243) | 5,875–8,813 (195–292) | Gamma | Case series and Standard Cost List for Health Economic evaluation in Thailand.[8,16] |
| ***Health Utilities (Quality of Life Estimates)*** | | | | |
| Hearing loss with or without VD | 0.58 | 0.34–0.81 | Beta | A multi-center, prospective study.[17] |
| Infective endocarditis (IE) | 0.67 | 0.40–0.94 | Beta | A single-center, prospective study.[18] |
| ***PALYs*** | | | | |
| Productivity index for no hearing loss | 0.96 | - | | A national longitudinal study.[19] |
| Productivity index for hearing loss | 0.95 | 0.72–0.96 | Uniform | A national longitudinal study.[19] |
| Productivity index for hearing loss with VD | 0.92 | 0.72–0.96 | Uniform | A cross-sectional survey.[20] |
| Productivity index for IE | 0.94 | 0.45–0.96 | Uniform | A coronary heart disease model development study.[21] |

Costs are presented as Thai baht ($US), Quarter 3, year 2019 values; CT, computed tomography; IE, infective endocarditis; IPD, inpatient department; PALYs, productivity-adjusted life years; THB, Thai baht; VD, vestibular dysfunction

Note

* Extrapolated from the recent 4-year data from 2015–2018 using logistic regression

** Direct medical care costs i.e. Medication, laboratory tests, X-ray, hospitalization

## Utilities and productivity indices

Country-specific health utility data was obtained from the European Quality of life five dimension (EQ-5D) index using the European visual analogue scale (VAS) value set for Thailand by

the EuroQol Group [25]. The EQ-5D is one of the most widely used multi attribute utility instruments for measuring health related quality of life, stratified into five dimensions: mobility, self-care, usual activities, pain/discomfort, and anxiety/depression [25]. The utility index is measured on a scale between 0 (complete equity in health) to 1 (complete inequity in health) [25]. In a similar way, productivity indices account for proportional reduction in work productivity, and were calculated based on presenteeism and absenteeism data reported by Nachtegaal et al. (2012) [19]. The study surveyed 1,295 working adults and compared the differences between sick leave taken by those with hearing impairment and their hearing able peers. The average amount of sick leave taken among those with insufficient and poor hearing ability was 4.3 days in four months (extrapolated to 12.9 days in one year), whereas the number of sick days taken by those who were in 'good' hearing test category was 3.1 days in four months (9.3 days per year). Findings from a cross sectional study of patients with bilateral and unilateral vestibular dysfunction by Sun et al (2014) [20] were used to estimate productivity loss attributable to hearing loss with vestibular dysfunction (productivity index = 0.9240). An assumption of 14 working days loss from IE was made based on the short-term disability benefits of 14 days leave among patients with coronary heart disease imputed in a model development study [21] as a conservative estimate. In the absence of the data concerning presenteeism in IE, only absenteeism was accounted in all productivity indices calculations for the sake of consistency. The equation with regards to productivity index calculation for each group is presented below [26]:

Productivity index = (full working year–absent days)/full working years

The number of working days in a years was estimated to be 250 days for full-time employment based on five working days per week and ten days of public holidays.

To estimate PALYs lived due to *S.suis* infection, each year lived in the labor force by the cohort was multiplied by a productivity index. We assumed that the economic value of each PALY was equivalent to annual gross domestic product (GDP) per worker, which equated to THB 436,039 ($US 13,955). This was derived from total Thailand GDP in 2018 (THB 16,318,033 million or $US 522,167.14) [27] divided by the estimated equivalent full-time (EFT) Thai workers in 2018 (n = 37,418,710) [28,29]. Owing to the scarcity of data on the part-time workforce participation and the predominance of full-time employment in Thailand, all employees were assumed to be in full-time employment.

## Sensitivity analyses

One-way sensitivity analyses were undertaken to assess the impact of uncertainty by varying key input parameters one at a time (Table 1). The ranges of probability of developing of meningitis with hearing loss and costs were set ±20% from the mean values. The upper and lower ranges for productivity indices were according to decreasing and increasing estimates of absenteeism. The working days lost from bilateral vestibular dysfunction was set as the lower bonds for productivity indices calculation for hearing loss with and without vestibular dysfunction [20], whereas the maximum working days lost (137 days) was applied for the productivity index for IE estimation [21]. The annual discount rates for both costs and benefits were varied to 0% and 6% [8].

We also performed probabilistic sensitivity analysis (PSA), to understand join uncertainty through Monte Carlo simulation using 10,000 iterations. The @RISK software version 8.0 for Excel was employed which allows multiple recalculations each time using a different set of random values for each parameter according to the defined distributions [30]. Gamma and beta distributions were applied for costs and utilities respectively as per decision modelling guidelines [31]. The uniform distribution was assumed for annual transition probabilities and

productivity indices. The 95% confidence interval (CI) of uncertainty ranges for the output variables attributable to the burden of disease were calculated.

## Results

The estimated number of Thai population affected by *S.suis* infection was 312 people (239 males and 73 females) per annum based on the country's population number in 2018 [29] and the 2019 annual incidence. There were more males infected than females (76.6% vs. 23.4%). The base-case results are presented in Table 2.

### Years of life lost to *S.suis* infection

The differences in the outputs of the two life tables (one each for the '*S.suis* infection cohort' and the hypothetical 'non *S.suis* infection cohort') represented the total years of life lost. The total discounted years of life lost to *S.suis* infection among the cohort was 769 (589 for males

**Table 2. Base-case analyses results, with uncertainty intervals.**

| | Male | | Female | | Total | | Difference (%) | Lost per person |
|---|---|---|---|---|---|---|---|---|
| | Infection | No infection | Infection | No infection | Infection | No infection | | |
| YLLs | 5,482 | 6,376 | 1,680 | 1,954 | 7,162 | 8,329 | 1,167 (14.01) | 3.74 |
| Disc 3% | 3,707 | 4,296 | 1,136 | 1,316 | 4,844 | 5,612 | 769 (13.71) | 2.46 |
| **Overall difference (uncertainty ranges 95% CI)** | | | | | | | 695 (12.38) to 841 (14.99) | |
| QALYs | 3,422 | 4,352 | 1,049 | 1,334 | 4,471 | 5,686 | 1,214 (21.37) | 3.89 |
| Disc 3% | 2,323 | 2,955 | 712 | 906 | 3,035 | 3,861 | 826 (21.39) | 2.64 |
| **Overall difference (uncertainty ranges 95% CI)** | | | | | | | 588 (15.23) to 1,098 (28.44) | |
| PALYs | 5,219 | 6,139 | 1,600 | 1,881 | 6,818 | 8,020 | 1,202 (14.98) | 3.85 |
| Disc 3% | 3,529 | 4,136 | 1,082 | 1,267 | 4,611 | 5,404 | 793 (14.67) | 2.54 |
| **Overall difference (uncertainty ranges 95% CI)** | | | | | | | 717 (13.27) to 867 (16.04) | |
| Total treatment cost ($US million) | 35,813,161 (1.17) | 0 | 10,949,289 (0.36) | 0 | 46,762,449 (1.52) | 0 | -46,762,449 (-1.52) | -149,661 (-0.005) |
| Disc 3% ($US million) | 34,804,660 (1.13) | 0 | 10,639,082 (0.35) | 0 | 45,443,742 (1.48) | 0 | -45,443,742 (-1.48) | -145,441 (-0.005) |
| **Overall difference (uncertainty ranges 95% CI)** | | | | | | | -55,222,281 to -38,966,206 (-$1,798,051 to -$1,268,749) | |
| Broader economic cost in million ($US million) | 2,276 (74.10) | 2,677 (87.16) | 697 (22.71) | 820 (26.71) | 2,973.29 (96.81) | 3,497 (113.87) | 524 (17.06) | 1.68 (0.55) |
| Disc 3% ($US million) | 1,539 (50.11) | 1,804 (58.73) | 472 (15.36) | 553 (18) | 2,011 (65.47) | 2,356 (76.73) | 346 (11.26) | 1.11 (0.36) |
| **Overall difference (uncertainty ranges 95% CI)** | | | | | | | 239,717,748 to 461,379,424 ($7,805,269 to $15,022,627) | |
| Net cost ($US million) | 2,2340 (72.93) | 2,677 (87.16) | 687 (22.35) | 820 (26.71) | 2,927 (95.29) | 3,497 (113.87) | 570,760,094 (18.58) | 1.83 (0.59) |
| Disc 3% ($US million) | 1,504 (48.98) | 1,804 (58.73) | 461 (15.01) | 553 (18) | 1,965 (63.99) | 2,356 (76.73) | 391,277,289 (12.74) | 1.25 (0.41) |
| **Overall difference (uncertainty ranges 95% CI)** | | | | | | | 281,879,015 to 509,098,159 ($9,178,050 to $16,576,361) | |

Costs are presented as Thai baht ($US), Quarter 3, year 2019 values; YLLs, years of life lived; Disc, discount rate; QALYs, quality-adjusted life years; PALYs, productivity-adjusted life years, CI, confidence interval

and 180 for females). At an individual level, this was equivalent 2.46 years of life lost per person (2.46 for males and 2.47 females).

## Quality-adjusted life years lost to *S.suis* infection

The total discounted QALYs lost to *S.suis* infection among the cohort was 826 (632 for males and 194 for females). Overall, at individual level, this was equivalent to 2.64 QALYs lost per (2.64 and 2.65 QALYs lost per male and female, respectively).

## Productivity-adjusted life years lost to *S.suis* infection

The total discounted PALYs lost to *S.suis* infection among the cohort was 793 (607 PALYs among males and 186 PALYs among females). Overall, 2.54 PALYs were estimated to be lost per person with *S.suis* infection (2.54 and 2.55 PALYs lost per male and female, respectively).

In terms of broader economic costs, the PALYs lost to *S.suis* equated to 346 million Thai baht (US$11.3 million) lost in GDP, which equated to an average of 1.1 million baht lost (US $36,033) GDP loss per person.

## Sensitivity analyses

Tornado diagrams were used to display one-way sensitivity analyses for QALYs and PALYs (Figs 2 and 3). The modelled results were most sensitive to productivity indices, utilities and discount rates.

Variation in discount rate caused the largest variation in estimated QALYs lost to *S.suis* infection, from -133.1% to 74.1%, followed by hearing loss utilities, which resulted in -74.9% to 74.9% change in QALY.

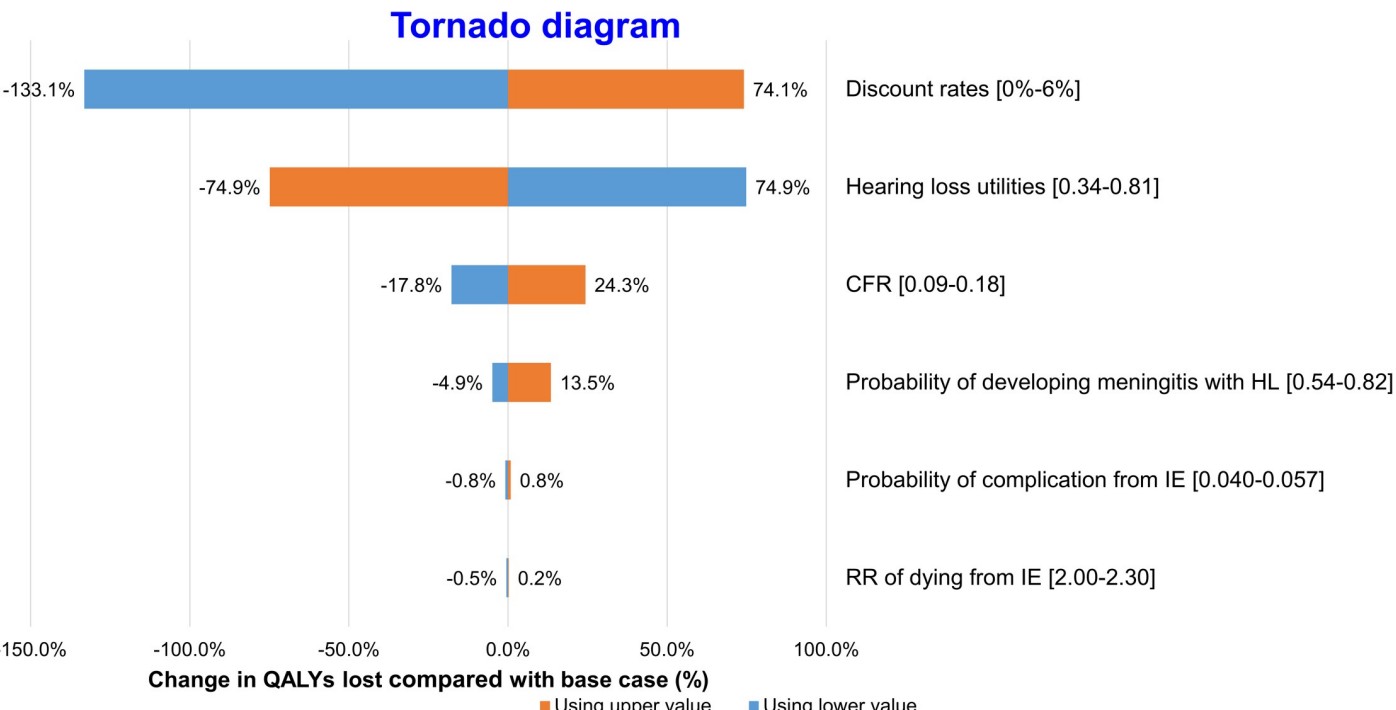

**Fig 2. One-way sensitivity analysis–QALYs.** Change in estimated QALYs lost to *S.suis* infection compared with base case (%)

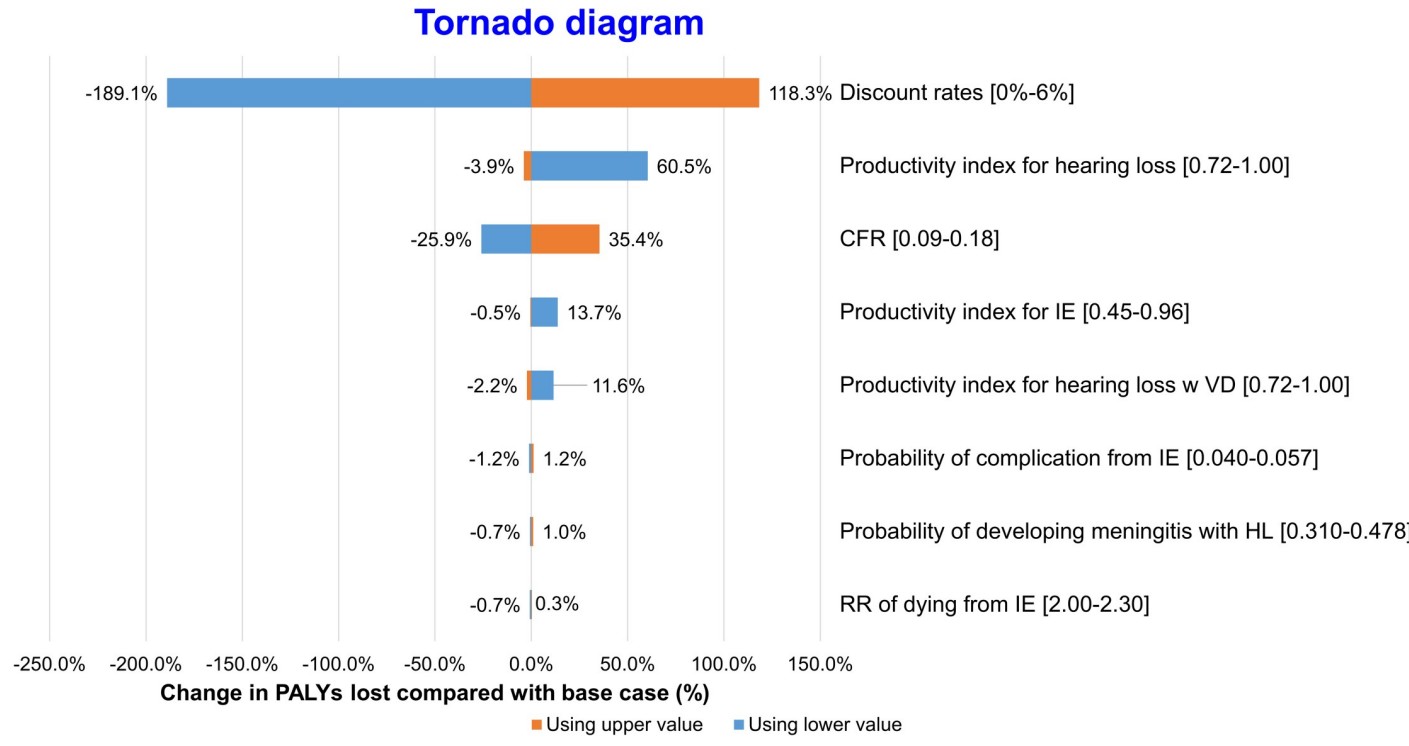

**Fig 3. One-way sensitivity analysis–PALYs.** Change in estimated PALYs lost to *S.suis* infection compared with base case (%)

Variation in discount rate also led to the greatest variation in estimated PALYs lost to *S.suis* infection, from -189.1% to 118.3%. Variation in case fatality and productivity index for hearing loss caused changes in PALYs lost from -25.9% to 35.4% and -3.9% to 60.5%, respectively. The other parameters exerted minimal impact on QALYs and PALYs lost.

The PSA indicates that the results were robust (Table 2). Overall burden due to *S.suis* infection ranged from 695 to 841 years of life, 588 to 1,098 QALYs, and 717 to 867 PALYs lost. The lost in GDP were estimated between 239.7 to 461.4 million Thai baht (US$ 7.8 to 15 million) whereas the total net costs varied between 281.9 to 509.1 million Thai baht (US$ 9.2 to 16.6 million) annually across the 10,000 simulations. Similarly, a slight variation in 95% uncertainty interval of total treatment cost was observed ranging between -55.2 to -39.0 million Thai baht (US$ 1.8 to 1.3 million).

## Discussion

The study highlights the impact of *S.suis* infection on the years of life, QALYs, and PALYs lived in Thailand. Among cohorts of Thai population with *S.suis* infection followed over a lifetime, it was predicted that the infection would lead to 769 years of life lost (14%), 826 QALYs lost (21.4%) and 793 PALYs lost (14.7%). This was equivalent to 2.46 years of life lost, 2.64 QALYs and 2.54 years productivity lost per person. A 14.7% loss of PALYs was also associated with a significant economic impact of 346 million Thai baht (US$11.3 million) loss in GDP. The years of life, QALYs, and PALYs lost were greater among men due to a higher prevalence of the disease in male population. The results are consistent with estimates from sensitivity analyses reflecting the robustness of the model.

*S.suis* infection causes a significant health and economic burden in Southeast Asia particularly Thailand and Vietnam. Meningitis is the most common clinical presentation which leads

to hearing loss and vestibular dysfunction among most survivors [1]. A relative high number of infective endocarditis (IE) as a major clinical manifestation was also reported in Thailand [11,32,33]. The disease affects mainly middle-age working men population [1,5]. The previous study in Vietnam showed a large burden of disease disproportionately distributed towards working-age men and substantial economic impact from *S.suis* corresponding 1,437 disability-adjusted life years (DALYs) lost with the annual direct cost US$370,000 in 2014 [4]. Despite the disease is largely under-reported, the number of cases are likely to increase. According to the recent report from Bureau of Epidemiology, Department of Disease Control, Ministry of Health, Thailand, there were 337 *S.suis* cases with 28 deaths from 1 January to 23 October 2019 [34] which is already more than the individuals with infection in 2019 predicted in our model.

Our study is the first to quantify the impact of *S.suis* infection in terms of PALYs, which is a novel measure that accounts for productivity loss attributable to the disease. PALYs is a relatively new concept depicting the disease impact on work productivity which has been introduced for only a few years [9,35,36]. PALYs are of same concept to QALYs (widely used in health technology assessments), but instead of adjusting years of life for time spent with reduced quality of life due to ill health, time spent with reduced work productivity is applied instead. QALYs are limited in capturing wider effects on economic impact from the disease on work productivity [37] which is an important perspective on the impact of ill-health. Therefore, PALYs may offer a more objective measure in terms of broader economy compared to QALYs, which may be seen as economic value loss to society. To our knowledge, there is only one previous study on economic impact of *S.suis* infection conducted in Vietnam [4]. However, the burden of disease was estimated in term of DALYs without estimation of long-term treatment such as follow-up audiological assessments and hearing aids and some major clinical manifestations and complications including infective endocarditis and cardiac complications after IE were not accounted. As the disease is regarded as being "rare" and largely under-reported, this information would provide important insights on the disease long-term outcome and economic consequences which is far beyond the acute infection to policy makers in planning for disease control and prevention.

Table 3 shows the cost estimates from *S.suis* infection compared to other infections in Thailand. Dengue affects a large proportion of the population (N = 81,000), equating to an annual loss of US$ 224 million [38]. In comparison, there are only 312 *S.suis* cases causing a loss of US $ 11.2 million annually (Table 3). However, the estimate loss per capita attributable to *S.suis* infection is far greater (US$ 36,033 per person) compared to dengue (US$ 5.03 per person) despite a nearly 260-fold difference in the number of annual cases.

When compared to pneumococcal disease, in terms of productivity impact, it is estimated that there are approximately 457,500 PALYs lost, which is equivalent to 0.15 PALYs lost per

**Table 3. Cost estimates from *S.suis* infection compared to other infections in Thailand.**

| Pathogen | Annual number of cases* | YLL loss | PALYs loss | | Broader economic cost ($2019)[a] | |
|---|---|---|---|---|---|---|
| | | | | Per case | Annual cost in $2019 million | Loss per person ($2019) |
| **Dengue [38]** | 81,000 | N.R. | N.R. | N.R. | 224 | 5.03 |
| **Pneumococcal disease[b][39]** | N.R. | 453,401 | 457,498 | 0.15 | 5,586 | N.R. |
| ***S.suis*** | 312 | 769 | 793 | 2.54 | 11.26 | 36,033 |

Costs are presented as Thai baht ($US), Quarter 3, year 2019 values; YLLs, years of life lived; PALYs, productivity-adjusted life years, CI, confidence interval

* The numbers were derived from different years

a Inflated to 2019 value

b 2018 value

N.R., Not report

person, whereas there are 793 PALYs lost or 2.54 PALYs lost per person attributable to *S.suis* infection. The estimated high per-case impacts of *S.suis* reflects its long-term sequelae among infected people which warrants attention from policy makers of this disproportionate individual burden, as averting a single case would reduce substantial economic burden.

The model was sensitive to productivity indices, utilities and discount rate which implies the wide ranges of disease sequelae, spectrum of clinical manifestations, and productivity attributed to the disease particularly hearing impairment with or without vestibular dysfunction. The lowest and highest ends of the 95% CI uncertainty range for YLLs, QALYs, PALYs and total net cost differed modestly from the base case analyses which suggest robustness of the model.

Food safety campaigns or public health interventions to raise the disease awareness are potentially effective for disease control and prevention. According to a study on impact of a food safety campaign in Phayao province in northern, Thailand, the disease incidence was markedly reduced during the first two years after the program implementation but started to rise again in the third year [40]. This emphasizes the existence of deep-rooted cultural behaviour of raw pork consumption which is a major cause of infection and the need for continuous effective public health interventions to improve awareness among Thai population and healthcare policy makers. Raw pork eating practice was also identified as one of significant predictors (raw pork eating, meningitis, and vestibular dysfunction) of *S.suis* hearing loss which is a major sequelae among most surviving patients [41]. Considering significant economic burden incurred from *S.suis* infection which is far more than acute infection treatment, reducing incidence of new cases can potentially save costs substantially in a long term.

There is a number of limitations from our study that warrant mention. Our direct medical cost estimate was based on the hospital charges at a tertiary hospital (Chiang Mai University Hospital) in Northern, Thailand. Therefore, this may not represent treatment cost in primary settings. However, *S.suis* infection is usually severe and life-threatening in which majority of cases would require tertiary care. The same utility for hearing loss and hearing loss with vestibular dysfunction was used in our model for base case estimate due to limited data available. However, even with highly conservative assumption, the results showed significant economic impact from *S.suis* infection. Some cost parameters including cochlear implant, rehabilitation relating to vestibular disorder were not included due to a lack of reliable information. According to the previous finding by Health Intervention and Technology Assessment Program (HiTAP), Thailand, cochlear implant is not a cost effective intervention in any group of patients with deafness [42]. The estimated cost per patient was as high as nearly 100,000 THB (US$ 31,811) at the first year and 30,000 THB (US$ 977) in subsequent years [43]. Therefore, it is unlikely that this intervention will be reimbursed in Thailand in the near future. Using life-table modeling which is a simple and commonly used tool, the age-specific mortality and RR of dying from IE would be constant over time which is a well-known limitation of the "life table assumption". Nonetheless, this approach was applied to both individuals with infection and no infection and the disease impact after the acute infection is unlikely to change tremendously. Therefore, this should not have resulted in significant change of our estimates and conclusion. In addition, local data was not available for some input parameters including productivity indices of *S.suis* infection in Thai population. In the absence of presenteeism data for IE, only absenteeism was used in the productivity indices calculation. This may have led to underestimation of productivity indices and economic impact. Finally, the simulated model assumed that GDP was stable rather than increase over time throughout the follow-up which is not likely to be the case. This would also have led to underestimation of the economic impact from *S.suis* infection. Notwithstanding with these limitations, the overall conclusion of the study would have unlikely changed.

In conclusion, *S.suis* infection imposes a significant economic burden both in terms of health and productivity. Future research to investigate the effectiveness of public health awareness programs and disease control interventions should be carried out to provide a clearer picture to assist decision making in public health strategies and resource allocations.

## Supporting information

**S1 Table. A data set of 133 *Streptococcus suis* patients admitted with total IPD cost** (DOCX)

## Acknowledgments

We would like to thank the staff at Chiang Mai University Hospital, Boonchira Kitisith, Ekkachai Jongkire, Areerat Kittikunakorn of the Office of Medical Records and Statistics, Chiang Mai University Hospital for their kind support in providing the acute treatment cost data for *S.suis* patients.

## Author Contributions

**Conceptualization:** Ajaree Rayanakorn, Danny Liew.

**Data curation:** Ajaree Rayanakorn, Zanfina Ademi, Danny Liew.

**Formal analysis:** Ajaree Rayanakorn, Zanfina Ademi, Danny Liew.

**Funding acquisition:** Learn-Han Lee.

**Investigation:** Ajaree Rayanakorn, Zanfina Ademi, Danny Liew.

**Methodology:** Ajaree Rayanakorn, Zanfina Ademi, Danny Liew.

**Project administration:** Ajaree Rayanakorn.

**Resources:** Ajaree Rayanakorn, Learn-Han Lee.

**Supervision:** Zanfina Ademi, Danny Liew, Learn-Han Lee.

**Writing – original draft:** Ajaree Rayanakorn.

**Writing – review & editing:** Ajaree Rayanakorn, Zanfina Ademi, Danny Liew, Learn-Han Lee.

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
