## [Decision Letter · Decision Letter 0]

8 Sep 2020

Dear Ms. Rayanakorn,

Thank you very much for submitting your manuscript "Burden of disease and productivity impact of Streptococcus suis infection in Thailand" for consideration at PLOS Neglected Tropical Diseases. As with all papers reviewed by the journal, your manuscript was reviewed by members of the editorial board and by several independent reviewers. In light of the reviews (below this email), we would like to invite the resubmission of a significantly-revised version that takes into account the reviewers' comments. Due to unexpected issued regarding the COVID-19 pandemic, we apologize for the long delay regarding the review of your manuscript.

We cannot make any decision about publication until we have seen the revised manuscript and your response to the reviewers' comments. Your revised manuscript is also likely to be sent to reviewers for further evaluation.

Sincerely,

Elsio Wunder Jr, D.V.M., Ph.D.

Deputy Editor

Reviewer's Responses to Questions

**Key Review Criteria Required for Acceptance?**

**Methods**

-Are the objectives of the study clearly articulated with a clear testable hypothesis stated?

-Is the study design appropriate to address the stated objectives?

-Is the population clearly described and appropriate for the hypothesis being tested?

-Is the sample size sufficient to ensure adequate power to address the hypothesis being tested?

-Were correct statistical analysis used to support conclusions?

-Are there concerns about ethical or regulatory requirements being met?

Reviewer #1: Minor revision: The introduction part is very short and most important limitations are missed that will support the claims by the authors. The motivation and gap of the study is not well defined. Regarding the method, the study design is properly meet the study objective. The target population is clearly defined.

Reviewer #2: 1. Table 1: Please indicate why the different distributions were chosen for different parameters in the model? (How did results change when all varied parameter distributions were set to uniform)?

2. Probability of dying after IE: A probability cannot be > 1 (it is stated as 2.2 in the table). Can you clarify?

3. IPD is not defined in the footnotes

4. Can the dollar values also be added to the cost parameters?

5. Is it reasonable to assume that the economic value of each PALY was equivalent to annual gross domestic product (GDP) per worker? Are there any data showing that persons exposed to and infected with S. suis might fall in a lower economic stratum?

Reviewer checklist:

-Are the objectives of the study clearly articulated with a clear testable hypothesis stated? Objectives are clear although there is no hypothesis-driven argument.

 -Is the study design appropriate to address the stated objectives? Yes

 -Is the population clearly described and appropriate for the hypothesis being tested? Yes

 -Is the sample size sufficient to ensure adequate power to address the hypothesis being tested? N/A

 -Were correct statistical analysis used to support conclusions? Yes.

 -Are there concerns about ethical or regulatory requirements being met? No.

**Results**

-Does the analysis presented match the analysis plan?

-Are the results clearly and completely presented?

-Are the figures (Tables, Images) of sufficient quality for clarity?

Reviewer #1: The authors present the results based on the years of life lost, QALY and Productivity Adjusted Life Years(PALYs). But the analysis seems confusing. The authors present the analysis for example, years of life lost estimated in terms of both numbers of infection or non-infection and the number is considered as years of life lost. This is not clear.

Reviewer #2: 1. It is perhaps not surprising that there is a cost to the economy and a reduction in productivity in a scenario where there is S. suis infection versus a scenario where there is no S. suis infection. The bigger questions are:

a. How does the impact of S suis infection compare with the impact of other neglected tropical diseases in the country: E.g.: intestinal protozoan infections, leishmaniasis, zoonotic malaria (Plasmodium knowlesi infection), Melioidosis, rickettsial infections, and leptospirosis (and others?): https://journals.plos.org/plosntds/article?id=10.1371/journal.pntd.0003575#:~:text=Intestinal%20protozoan%20infections%20are%20widespread,selected%20rickettsial%20infections%2C%20and%20leptospirosis. “Neglected Tropical Diseases among the Association of Southeast Asian Nations (ASEAN): Overview and Update”

b. What interventions are there to address S. suis infection and what is the cost-effectiveness of this intervention. 

While I understand that the second question is beyond the scope of this paper, can you add a table summarizing the impact of the other NTDs? Is $11 million loss a big or small impact in comparison to other NTDs and health threats?

How much is currently invested in S suis control?

Reviewer checklist:

-Does the analysis presented match the analysis plan? Yes.

 -Are the results clearly and completely presented? Yes. Although as noted above, the paper's value would be very much improved if the findings were placed in context.

 -Are the figures (Tables, Images) of sufficient quality for clarity? Yes, although one additional figure, at least one additional table to place the findings in context (i.e. compare with other NTDs and health threats), and some table edits are suggested above.

**Conclusions**

-Are the conclusions supported by the data presented?

-Are the limitations of analysis clearly described?

-Do the authors discuss how these data can be helpful to advance our understanding of the topic under study?

-Is public health relevance addressed?

Reviewer #1: The conclusions supported by the data and the authors discussed the limitations on their analysis clearly.

Reviewer #2: 1. Can the authors do a more convincing job convincing the reader that “S.suis infection imposes a significant economic burden both in terms of health and productivity”? Thailand has a GDP of over $540 Billion (https://tradingeconomics.com/thailand/gdp). The $11.3 million loss from the annual GDP is 0.002% loss of GDP which seems small. How does this compare to other NTDs and other health-related causes of GDP loss? 

2. Is there any current investment in S. suis prevention? What is the annual cost of that prevention program currently or in the past?

3. Given that PALY is a new measure, can the authors indicate why it is important for this paper. If no other paper has reported a PALY, how can authors compare the PALY of S. suis with other NTDs or health threats? Do the authors think this should be a standard measure for future economic impact assessments? There is not much discussion of the value of PALY in the discussion section given its novelty.

Reviewer checklist:

-Are the conclusions supported by the data presented? No, for the reason noted in #1 above.

 -Are the limitations of analysis clearly described? Yes.

 -Do the authors discuss how these data can be helpful to advance our understanding of the topic under study? Partly. See comments #1 and #2 above.

 -Is public health relevance addressed? Partly. See comments #1 and #2 above.

**Editorial and Data Presentation Modifications?**

Reviewer #1: Minor Revision

Reviewer #2: These comments have been covered above.

**Summary and General Comments**

Reviewer #1: The authors have addressed an important question on the burden of disease and productivity impact of Streptococcus-suis infection in Thailand. They estimated the health and economic burden of S.suis infection in terms of years of life lost, quality-adjusted life years (QALYs) lost, and productivity-adjusted life years (PALYs) lost and concluded that the infection have burden on health and producativity. After reading the manuscript I have the following minor comments.

The introduction part is very short and most important limitations are missed that will support the claims by the authors. The motivation and gap of the study is not well defined. Regarding the method, the study design is properly meet the study objective. The target population is clearly defined.

The authors present the results based on the years of life lost, QALY and Productivity Adjusted Life Years(PALYs). But the analysis seems confusing. The authors present the analysis for example for years of life lost estimated interms of both numbers of infection or non-infection and the number is considered as years of life lost. This is not clear.

Figure captions are not included in the text of the manuscript.

Reviewer #2: As noted in the results section, It is perhaps not surprising that there is a cost to the economy and a reduction in productivity in a scenario where there is S. suis infection versus a scenario where there is no S. suis infection. The bigger questions are:

a. How does the impact of S suis infection compare with the impact of other neglected tropical diseases in the country: E.g.: intestinal protozoan infections, leishmaniasis, zoonotic malaria (Plasmodium knowlesi infection), Melioidosis, rickettsial infections, and leptospirosis (and others?): https://journals.plos.org/plosntds/article?id=10.1371/journal.pntd.0003575#:~:text=Intestinal%20protozoan%20infections%20are%20widespread,selected%20rickettsial%20infections%2C%20and%20leptospirosis. “Neglected Tropical Diseases among the Association of Southeast Asian Nations (ASEAN): Overview and Update”

b. What interventions are there to address S. suis infection and what is the cost-effectiveness of this intervention. 

Is $11 million loss a big or small impact compared with other NTDs and health threats? The paper needs additional tables that allow comparison with other NTDs and health threats to place the findings of the paper in context.

PLOS authors have the option to publish the peer review history of their article (what does this mean?). If published, this will include your full peer review and any attached files.

Reviewer #1: No

Reviewer #2: No
---

## [Decision Letter · Decision Letter 1]

12 Nov 2020

Dear Dr. Rayanakorn,

We are pleased to inform you that your manuscript 'Burden of disease and productivity impact of Streptococcus suis infection in Thailand' has been provisionally accepted for publication in PLOS Neglected Tropical Diseases.

Best regards,

Elsio Wunder Jr, D.V.M., Ph.D.

Deputy Editor

Elsio Wunder Jr

Deputy Editor

Reviewer's Responses to Questions

**Key Review Criteria Required for Acceptance?**

**Methods**

-Are the objectives of the study clearly articulated with a clear testable hypothesis stated?

-Is the study design appropriate to address the stated objectives?

-Is the population clearly described and appropriate for the hypothesis being tested?

-Is the sample size sufficient to ensure adequate power to address the hypothesis being tested?

-Were correct statistical analysis used to support conclusions?

-Are there concerns about ethical or regulatory requirements being met?

Reviewer #2: -Are the objectives of the study clearly articulated with a clear testable hypothesis stated? Yes

-Is the study design appropriate to address the stated objectives? Yes

-Is the population clearly described and appropriate for the hypothesis being tested? Yes

-Is the sample size sufficient to ensure adequate power to address the hypothesis being tested? N/A

-Were correct statistical analysis used to support conclusions? Yes

-Are there concerns about ethical or regulatory requirements being met? Yes

**Results**

-Does the analysis presented match the analysis plan?

-Are the results clearly and completely presented?

-Are the figures (Tables, Images) of sufficient quality for clarity?

Reviewer #2: -Does the analysis presented match the analysis plan? Yes

-Are the results clearly and completely presented? Yes

-Are the figures (Tables, Images) of sufficient quality for clarity? Yes

**Conclusions**

-Are the conclusions supported by the data presented?

-Are the limitations of analysis clearly described?

-Do the authors discuss how these data can be helpful to advance our understanding of the topic under study?

-Is public health relevance addressed?

Reviewer #2: -Are the conclusions supported by the data presented? Yes

-Are the limitations of analysis clearly described? Yes

-Do the authors discuss how these data can be helpful to advance our understanding of the topic under study? Yes

-Is public health relevance addressed? Yes

**Editorial and Data Presentation Modifications?**

Reviewer #2: No additional changes are recommended

**Summary and General Comments**

Reviewer #2: The authors have addressed the questions raised previously. I have no further comments or questions.

PLOS authors have the option to publish the peer review history of their article (what does this mean?). If published, this will include your full peer review and any attached files.

Reviewer #2: No

---

## [Editor Report · Acceptance letter]

19 Jan 2021

Dear Dr. Rayanakorn,

We are delighted to inform you that your manuscript, "Burden of disease and productivity impact of Streptococcus suis infection in Thailand," has been formally accepted for publication in PLOS Neglected Tropical Diseases.

Best regards,

Shaden Kamhawi

co-Editor-in-Chief

Paul Brindley

co-Editor-in-Chief
